# On the Stochastic Mechanics Foundation of Quantum Mechanics

**Michael Beyer** *,† and **Wolfgang Paul** †

Institut für Physik, Martin-Luther-Universität Halle-Wittenberg, 06099 Halle (Saale), Germany;
wolfgang.paul@physik.uni-halle.de
* Correspondence: michael.beyer@physik.uni-halle.de
† These authors contributed equally to this work.

**Abstract:** Among the famous formulations of quantum mechanics, the stochastic picture developed since the middle of the last century remains one of the less known ones. It is possible to describe quantum mechanical systems with kinetic equations of motion in configuration space based on conservative diffusion processes. This leads to the representation of physical observables through stochastic processes instead of self-adjoint operators. The mathematical foundations of this approach were laid by Edward Nelson in 1966. It allows a different perspective on quantum phenomena without necessarily using the wave-function. This article recaps the development of stochastic mechanics with a focus on variational and extremal principles. Furthermore, based on recent developments of optimal control theory, the derivation of generalized canonical equations of motion for quantum systems within the stochastic picture are discussed. These so-called quantum Hamilton equations add another layer to the different formalisms from classical mechanics that find their counterpart in quantum mechanics.

**Keywords:** stochastic mechanics; quantum mechanics; stochastic foundation of quantum mechanics; stochastic differential equations

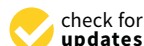



## 1. Introduction

"Shut up and calculate!" We probably all know this quote by David Mermin [1] about his time as a young researcher at Havard University. The next sentence is seldom quoted: "But I won't shut up". Both refer, of course, to our relationship as physicists with quantum mechanics. We mostly agree on how to calculate things, i.e., for non-relativistic problems, we use the Schrödinger equation, but when it comes to deciding what the theory means, we are facing a zoo of suggestions and opinions. Why is this possible in an exact science like physics? Mermin's complete quote tells us that it is not sufficient to be able to calculate; as physicists, we are required to understand and we want to understand.

We think that the difficulty in formulating a complete and consistent physical picture of quantum phenomena lies mainly in the fact that the Schrödinger equation alone does not offer sufficient mathematical structure to determine the correct physical picture of the phenomena it describes. Imagine having the Hamilton–Jacobi equation as the only tool of classical analytical mechanics. The famous cannon ball on its flight from the freshman physics course would then be described by two fields, the probability density $\rho(x(t), t)$ and the action $S(x(t), t)$, extending from here to the Orion nebula (if we factor in positional measurement errors). Once the ball hits the target, these functions would collapse into the impact position. We would assume a complementary of particle and field descriptions and a special role of the measurement process. This is a world view we are taught in the classroom for quantum mechanics, but would never accept for classical mechanics.

One of the origins of the belief that the Schrödinger equation offers **the** and not **a** complete description of quantum phenomena lies in the no hidden variable theorem first formulated by John von Neumann [2]. The proof was shown to be wrong shortly after publication by the young German mathematician Grete Hermann [3], but this went largely

unrecognized. After David Bohm [4] had given a constructive proof of the possibility of a hidden variable theory in the 1950s, John Bell rediscovered the fault in von Neumann's proof and started to formulate his famous inequalities [5]. The question about the possibility or not of a hidden variable formulation of quantum mechanics has undergone much development since then, and discussing it is outside the scope of this article. Today, one can perhaps summarize the current state of the discussion by saying that the Bell inequalities and related theorems imply that measured observables are contextual, i.e., a counterfactual determinism postulating that a property also had a certain value determined by a specific experimental setup in the past, before the measurement was taken, is excluded.

Nelson's stochastic mechanics formulation of quantum mechanics, which started its development with his article *On the derivation of the Schrödinger equation from Newtonian mechanics* in 1966 [6], is completely in accord with the requirements of Bells inequalities. By now, it has been developed to a mathematical rigor that completely parallels the formulation of classical analytical mechanics. It thus provides sufficient mathematical structure to suggest a clear physical picture of quantum phenomena. We will discuss this mathematical structure, which we would like to call quantum analytical mechanics, in the rest of this review.

Section 2 presents Nelson's original idea and a synopsis of the different ideas built on that. In Section 3, we will focus especially on the role of variational principles in stochastic mechanics, while Section 4 presents the derivation and application of quantum Hamilton equations. Finally, Section 5 will present an outlook on future developments.

## 2. Stochastic Mechanics

There is an often implicit assumption in our typical approach as physicists to model a natural phenomenon: we identify a system, modeled by its Hamiltonian, $H_s(r)$, its environment, $H_e(R)$, and their interaction, $H_i(r, R)$. We then assume that the interaction with the environment is small, $H_i(r, R) \ll H_s(r)$, and can be treated perturbatively, so that the model can be reduced to $H_s(r)$ to understand the main properties of the problem.

Nelson's starting point in his 1966 paper can be described as stating that this does not work in the quantum world. His *assumption of universal Brownian motion* means that the interaction of a quantum particle with the universally present background radiation is not a small perturbation, but essential to the behavior of a quantum system. This interaction is unknowable in detail, but only its statistical properties enter, as typical for Brownian motion. There are three physical assumptions defining the theory:

1.　The path of a quantum particle is a realization of a conservative diffusion. It is driven by Brownian motion and may be written as an Itô stochastic differential Equation (SDE)

$$\mathrm{d}X(t) = b_f(X(t), t)\mathrm{d}t + \sigma \mathrm{d}W_f(t). \tag{1}$$

For notational simplicity, we discuss only a one-dimensional motion here. In this equation, $b_f(x, t)$ is the forward drift depending on the current position $x = X(t)$, $\sigma$ the square root of the diffusion coefficient and $\mathrm{d}W_f(t)$ a forward Wiener process, i.e., a process with Gaussian increments with mean zero and width $\mathrm{d}t$, which is independent of the future, i.e., of all $X(s), s > t$. Note that capital letters, e.g., $X(t)$ and $X = (X(t))_{t \in [t_0, t_1]}$, indicate random variables and the corresponding stochastic processes, whereas the small letters refer to their values. A conservative diffusion is non-dissipative and thus time reversible [7], where the backward in time process exists

$$\mathrm{d}X(t) = b_b(X(t), t)\mathrm{d}t + \sigma \mathrm{d}W_b(t). \tag{2}$$

with increments $\mathrm{d}W_b(t)$, which are independent of the past, i.e., all $X(s), s < t$.

2.　The diffusion coefficient should vanish for macroscopic objects, so one can assume that it is inversely proportional to the mass of the particle. The proportionality constant then has the units of action, and, in view of a later identification, one can write $\sigma^2 = \hbar/m$. In fact, one can show that the interaction of a Brownian particle

with the background radiation gives rise to a temperature-independent diffusion coefficient of this magnitude [8].

3. One knows that the solutions $X(t)$ of the above equations are with probability one everywhere continuous, but nowhere differentiable. So, how does one define velocity or acceleration? Nelson suggested an average forward and backward differential

$$
\begin{aligned}
D_f X(t) &= \lim_{\Delta t \to 0} \mathrm{E}_t \left[ \frac{X(t+\Delta t) - X(t)}{\Delta t} \right] \\
D_b X(t) &= \lim_{\Delta t \to 0} \mathrm{E}_t \left[ \frac{X(t) - X(t-\Delta t)}{\Delta t} \right]
\end{aligned}
\tag{3}
$$

where the capital $\mathrm{E}_t$ denotes the expectation conditional on $X(t) = x$. For differentiable curves $D_f X(t) = D_b X(t) = \dot{x}(t) = v(t)$, the velocity of the particle. With this, Nelson postulated the validity of a stochastic Newton law for the stochastic acceleration

$$
ma(X(t)) = m \frac{1}{2} \left( D_f D_b + D_b D_f \right) X(t) = F(X(t)) .
\tag{4}
$$

As in classical analytical mechanics, this should amount to a complete description of the motion, so one should be able to derive a Lagrangian, Hamiltonian and Hamilton–Jacobi formulation of quantum motion based on these three physical assumptions. The Hamilton–Jacobi formulation of quantum motion as described by stochastic mechanics is the Schrödinger equation, the goal with which Schrödinger set out to derive his equation [9].

The mean forward and backward derivatives of the particle position just give the drift coefficients in the forward and backward equations

$$
D_f X(t) = b_f(X(t), t) \quad D_b X(t) = b_b(X(t), t) .
\tag{5}
$$

On the level of probability distributions, the forward and backward stochastic differential equations are equivalent to two Fokker–Planck equations:

$$
\frac{\partial}{\partial t} \rho(x, t) = -\frac{\partial}{\partial x} \left[ b_f(x, t) \rho(x, t) \right] + \frac{\hbar}{2m} \frac{\partial^2}{\partial x^2} \rho(x, t)
\tag{6}
$$

$$
\frac{\partial}{\partial t} \rho(x, t) = -\frac{\partial}{\partial x} [b_b(x, t) \rho(x, t)] - \frac{\hbar}{2m} \frac{\partial^2}{\partial x^2} \rho(x, t) .
\tag{7}
$$

The latter equation carries a minus sign in front of the diffusion coefficient, which is due to the time reversal. The sum of the two Fokker–Planck equations gives the continuity equation

$$
\frac{\partial}{\partial t} \rho(x, t) + \frac{\partial}{\partial x} [v(x, t) \rho(x, t)] = 0 ,
\tag{8}
$$

with the current velocity $v(x, t) = (b_f(x, t) + b_b(x, t))/2$. The difference of the forward and backward drifts $u(x, t) = (b_f(x, t) - b_b(x, t))/2$ defines the osmotic velocity. Subtracting the two Fokker–Planck equations yields

$$
u(x, t) = \frac{\hbar}{m} \frac{\partial}{\partial x} \ln[\rho(x, t)] = \frac{\hbar}{2m} \frac{\partial}{\partial x} R(x, t) ,
\tag{9}
$$

where we wrote the probability as $\rho(x, t) = \exp\{2R(x, t)\}$. The two coupled forward-backward stochastic differential equations for the position process thus read

$$
\begin{aligned}
\mathrm{d}X(t) &= (v(X(t), t) + u(X(t), t))\mathrm{d}t + \sigma \mathrm{d}W_f(t) \\
\mathrm{d}X(t) &= (v(X(t), t) - u(X(t), t))\mathrm{d}t + \sigma \mathrm{d}W_b(t) .
\end{aligned}
\tag{10}
$$

From [10], it follows that the current velocity is curl-free (in more than one dimension), so we can define another scalar field $S(x, t)$ by

$$v(x,t) = \frac{1}{m}\frac{\partial}{\partial x}S(x,t) \, . \tag{11}$$

The analogy to classical analytical mechanics suggests the prefactor and identifies $S(x,t)$ as the action. To derive partial differential equations for $R(x,t)$ and $S(x,t)$, we use the Itô formula

$$\mathrm{d}f(X(t),t) = \frac{\partial}{\partial t}f(X(t),t)\mathrm{d}t + \frac{\partial}{\partial x}f(X(t),t)\mathrm{d}X(t) + \frac{1}{2}\frac{\partial^2}{\partial x^2}f(X(t),t)(\mathrm{d}X(t))^2 \tag{12}$$

where the differentials are evaluated up to linear order in $\mathrm{d}t$. We also assume a conservative force field $F(x) = -\mathrm{d}U/\mathrm{d}x$ to obtain (for a more detailed account see [6,7,11])

$$\frac{\partial R}{\partial t} + \frac{1}{2m}\frac{\partial^2 S}{\partial x^2} + \frac{1}{m}\frac{\partial R}{\partial x}\frac{\partial S}{\partial x} = 0 \tag{13}$$

$$\frac{\partial S}{\partial t} + U + \frac{1}{2m}\left(\frac{\partial S}{\partial x}\right)^2 - \frac{\hbar^2}{2m}\left[\left(\frac{\partial R}{\partial x}\right)^2 + \frac{\partial^2 R}{\partial x^2}\right] = 0. \tag{14}$$

These equations are the Madelung [12] equations, the stochastic mechanics' counterpart to the Hamilton–Jacobi equations in classical mechanics, which constitute the hydrodynamic formulation of Newtonian mechanics. They differ from the classical equations by the additional terms depending on the strength $\hbar/m$ of the stochastic forces. For $\hbar/m \to 0$, the classical Hamilton–Jacobi equations are exactly recovered. The first of these two equations is nothing but the conservation of probability in another form, the second is the momentum balance.

Identifying $\psi(x,t) = \exp\{R(x,t) + \frac{i}{\hbar}S(x,t)\}$ one sees that $\psi$ solves the Schrödinger equation for solutions $R$ and $S$ of the Madelung equations. However, only for node free $\psi(x,t)$, i.e., in the ground state of a quantum problem, there is true equivalence. It will become clearer in the discussion of the variational principles in Section 3 how this limitation to the ground state comes about. The limited equivalence of the Madelung equations to the Schrödinger equation directly raised the questions about the existence of the Nelson diffusions for excited states and the behavior around nodes. Carlen [13] could show that such diffusion processes also exist for the excited states and called them singular diffusions. For highly excited states in the hydrogen atom, it was possible based on stochastic mechanics to show that they approach Keplerian orbits [14], which had been expected based on Bohr's correspondence principle.

Rather early on, different versions of variational principles have been formulated [10,15–17]. These will be discussed in detail in the next section. An important ingredient to formulate them on general Riemannian manifolds was provided by the analysis of stochastic parallel displacements [18]. A particular case of diffusion on a Riemannian manifold occurs in the description of spin, where the orientation variable assumes values on the coordinate manifold of SO(3). Dankel transferred Nelson's approach to this case based on the description of spinning tops and the Bopp–Haag Hamiltonian [19] (see also the discussion by Faris [20]). While this description works with a continuous variation of orientations, there have also been attempts to formulate the stochastic mechanics of spins assuming the quantization into discrete values from the outset [21].

The physical picture of (point) particles following continuous paths whose statistics is governed by the Madelung equations or the Schrödinger equation, respectively, is rather intuitive. Feynman's path integral approach is a functional integral formulation of this path statistics [22]. When we accept these paths as a physical reality, the interpretational problems associated with the Copenhagen interpretation of quantum mechanics do not occur from the outset [23]. Nevertheless, one has to and can formulate a stochastic mechanics description of the measurement process [24].

The formulations of stochastic mechanics discussed above are all operationalized by solving the Schrödinger equation in the first step and then using the stochastic mechanics' background to calculate further properties based on this solution. This has led to interesting numerical applications, where the sample paths of the conservative diffusions described by the wave function were generated and analyzed [25–30]. This turned out to be especially fruitful when one asks about the time some quantum processes take. There is no time operator in quantum mechanics, so there is no direct operational definition for the calculation of the duration of a process as the expectation value of some self-adjoint operator. For the underlying diffusion processes, it is quite natural to ask, e.g., how long it takes to traverse a barrier [31] or to diffuse from the plane of a double-slit to the measurement screen [29]. We will come back to the problem of barrier traversal, aka tunneling times in Section 4.2.

We will discuss the formulation of stochastic mechanics as a stochastic optimal control problem in the next section. This also allows to derive quantum Hamilton equations [32, 33], which allow for a direct (mostly only numerical) solution without recourse to the Schrödinger equation. By construction, this proves that the Schrödinger equation is not **the**, but only **a** complete formulation of quantum mechanics, contrary to the claims of the Copenhagen interpretation. In Section 4.1, we will discuss an example for this way to solve a quantum problem.

### 3. Variational Principles in Stochastic Mechanics

The stochastic formalism introduced by Nelson relies in particular on a postulated, generalized Newton law for the mean acceleration of a particle (4). Clearly, in the classical limit, the expectation is not needed, and thus Newton's 2nd law follows. From classical mechanics, it is well known that these dynamic differential equations can be reformulated in various forms, such as integral equations with Hamilton's principle. This principle of stationary action states that the dynamics of the system are set by a functional's extremization, i.e., the action

$$S[x] = \int_{t_0}^{t_1} \mathcal{L}(x, \dot{x}, t)\mathrm{d}t \tag{15}$$

is stationary with respect to a critical path $x = (x(t))_{t \in [t_0, t_1]}$ with fixed end points $x(t_0)$, $x(t_1)$, where the Lagrangian $\mathcal{L}(x, \dot{x}, t) = \frac{m}{2}\dot{x}^2 - V(x, t)$ contains the physical information of the system. Here and further on, short notations may be used, e.g., $x = x(t)$, $\dot{x} = \dot{x}(t)$, $X = X(t)$. A vanishing functional derivative, i.e., $\delta_x S[x]|_{x=x^*} = 0$, leads to the Euler–Lagrange equations:

$$\frac{\partial \mathcal{L}}{\partial x^*} - \frac{\mathrm{d}}{\mathrm{d}t}\frac{\partial \mathcal{L}}{\dot{x}^*} = 0. \tag{16}$$

Variational principles or, related to that, optimal control problems in the stochastic picture, where, generally, one searches criticality for non-differentiable paths, should include the classical variational principles as special cases. The criticality with respect to a cost functional can certainly not control the path itself due to the noise term. However, it is possible to adjust the mean of the stochastic behavior, namely the expectation value. In the stochastic mechanics' framework, two major suggestions were put forward by Yasue [15,16] and Guerra and Morato [10] in the 80s. The important aspect here is the time-reversibility of the diffusion, which means there is a forward and backward diffusion, and secondly the derivatives are generalized to the smoothed mean forward and backward derivatives which Nelson introduced. The 'Lagrangian' approach by Yasue considers a cost functional

$$S[X] = \mathrm{E}\left[\int_{t_0}^{t_1} \mathcal{L}(X, b_f, b_b, t)\mathrm{d}t\right], \tag{17}$$

where E denotes expectation with respect to the distribution of the stochastic process $X$ and in the suitably defined Lagrangian $\mathcal{L}$ the velocity is generalized by the two conditional smooth velocities $D_f X(t) = b_f(X(t), t), D_b X(t) = b_b(X(t), t)$ with respect to the present. The variation of the functional around a critical $X^*$, i.e., a deviation from the critical process $X^*$ by a stochastic process $Z$, has to fulfill $S[X^* + Z] - S[X^*] = \mathcal{O}(|Z|)$, with fixed endpoints $X(t_0)$, $X(t_1)$ yields stochastic Euler–Lagrange equations

$$\frac{\partial \mathcal{L}}{\partial X} - D_f \left( \frac{\partial \mathcal{L}}{\partial D_b X} \right) - D_b \left( \frac{\partial \mathcal{L}}{\partial D_f X} \right) = 0 \,. \tag{18}$$

These resemble the Euler–Lagrange equations in the deterministic case, where $D_f X = D_b X = \dot{x}$. The choice of $\mathcal{L} = \mathcal{L}_Y = \frac{1}{4}((D_f X)^2 + (D_b X)^2) - V$ for the Lagrangian leads to the Nelson–Newton law (4) for (18). Written in terms of the current and osmotic velocities, the Lagrangian, in that case, resembles the classical one

$$\mathcal{L}_Y = T - V = \frac{m}{2}(v^2 + u^2) - V \,. \tag{19}$$

Different from the 'Lagrangian' approach by Yasue using stochastic calculus, Guerra and Morato [10] use stochastic control theory. The cost functional is to be optimized w. r. t. to the smoothed (forward) velocity $b_f(X(t), t) = D_f X(t)$ (works similarly for $D_b X(t)$) subject to the control equation $\mathrm{d}X(t) = b_f(X(t), t)\mathrm{d}t + \sigma \mathrm{d}W_f(t)$ and fixed initial $\rho(\cdot, t_0)$ and final probabilities $\rho(\cdot, t_1)$. Thus, this approach is based on the fluid dynamics picture. Their Lagrangian is defined as

$$\mathcal{L}_G = \mathcal{L}(X, D_f X, D_b X, t) = \frac{m}{2} D_f X(t) \cdot D_b X(t) - V(X(t), t) \,, \tag{20}$$

where the variation of the cost functional w. r. t. a deviation from the critical velocity corresponds here to a variation of the critical drift $b_f^*$ by a stochastic process $Z$. Again, the quantum Hamilton–Jacobi like Equation (14) for the velocities $v = \frac{1}{2}(D_f X + D_b X)$ and $u = \frac{1}{2}(D_f X - D_b X)$ are derived. Comparing these two definitions of the Lagrangians $\mathcal{L}_G$, $\mathcal{L}_Y$, they differ by a sign in front of the osmotic energy

$$\mathcal{L}_G = \frac{m}{2}(v^2 - u^2) - V \,. \tag{21}$$

The special role of this additional (kinetic) term can be explained by taking the expectation and by using (9)

$$\mathrm{E}\left[ \frac{m}{2} u^2(x, t) \right] = \int \left( \frac{\hbar}{2m} \frac{\nabla^2 \rho(x, t)}{\rho(x, t)} \right) \rho(x, t) \mathrm{d}x = \mathrm{E}[V_Q] \,. \tag{22}$$

Thus, under expectation, the osmotic energy equals Bohm's postulated quantum potential $V_Q$ in the pilot-wave theory [4]. From that point of view, the fluctuation of the kinetic energy with the minus sign $-\frac{m}{2}u^2$ in (21) could be interpreted as an additional contribution to the potential $V$ without the need to postulate a non-local potential. Furthermore, in (22) appears the gradient of the so-called Fisher information (functional) that is used, for example, in the context of optimal mass transport theory. There it was shown that the nonrelativistic evolution of quantum systems, namely the Madelung fluid equations, can be deduced from the optimal interpolation of flows in the so-called Wasserstein space between fixed initial and final measures [34]. Hence, the probability distribution is varied. In [35], the sign of the dissipation term, and with that the sign of the osmotic energy in the Lagrangian, is explained by the underlying picture, namely in the particle formulation (Yasue) or the fluid dynamic (Guerra) formulation.

All of the suggested variational principles recover the Hamilton–Jacobi like equation, including the quantum correction terms, which is one of the Madelung equations. The second equation, namely the continuity equation, is satisfied due to the assumption of time-reversibility. In [32], Pavon introduced the so-called quantum Hamilton principle based on two variational principles recovering both of the Madelung equations. For that, it is proposed to search for saddle-points for the current **and** osmotic velocity, i.e., two controls, with the help of Lagrangian functionals. As shown in (21), the Lagrangian is convex in $v$, while it is concave in $u$. Pavon suggests using this Lagrangian and considers it as a zero-sum stochastic differential game for two players, where the player controlling the current velocity attempts to minimize the cost, whereas the one controlling the osmotic part tries to maximize it,

$$
J_1[X^*, u^*, v^*] = \max_X \min_v \max_u E\left[\int_{t_0}^{t_1} \mathcal{L}_G(X, u, v, t)\mathrm{d}t + S_1(X(t_1))\right]. \tag{23}
$$

Here, small letters for the stochastic controls $v(t)$ and $u(t)$ are used and $S_1(\cdot)$ is a given continuous function as final constraint. Instead of final conditions, initial conditions could also be used. Additionally, a second variational principle based on the systems entropy is postulated based on the configurational entropy of the system $S_E(t) = \int -\rho(x, t) \ln \rho(x, t)\mathrm{d}x$ and seeks to increase the entropy in the diffusion process,

$$
J_2[X^*, u^*, v^*] = \max_X \max_v \min_u E\left[\int_{t_0}^{t_1} \left[-\sigma^{-2} v^T u\right]\mathrm{d}t + R_1(X(t_1))\right], \tag{24}
$$

where $R_1(\cdot)$ is a continuous given function. This so-called saddle-point entropy production principle corresponds to the continuity equation and thus to a time-reversible diffusion. Using complex numbers, both variational principles can be written as one principle, which Pavon called the quantum Hamilton principle. This was reformulated as a stochastic optimal control problem [33] based on the mathematical theory developed in the last decades [36,37] with optimal feedback controls for Nelson's diffusion processes. This subsequently allows to derive the stochastic Hamilton equations by analogy to the deterministic optimal control problem where a stochastic Hamiltonian is pointwise extremized. This is considered in the next chapter.

## 4. Quantum Hamilton Equations

Hamilton's principle in classical mechanics can be reformulated as an optimal control problem by introducing the control $v(t)$ that minimizes the action $S[v]$ under a controlled equation

$$
\begin{aligned}
S[v] &= \int_{t_0}^{t_1} \mathcal{L}(x, v, t)\mathrm{d}t\,, \\
\dot{x}(t) &= v(t)\,, \quad x(t_0) = x_{t_0}\,.
\end{aligned} \tag{25}
$$

Pontryagin's maximum principle [38] then states that the tuple of optimal state trajectory $x^*$, optimal control $v^*$ and associated costate $p^*$ have to pointwise maximize the associated Hamiltonian $\mathcal{H}(x, v, p, t) = p^T(t)v(t) - \mathcal{L}(x, v, t)$ for all admissible controls $v$ in $t \in [t_0, t_1]$, so that the following holds

$$
\begin{aligned}
\partial_v \mathcal{H}(x^*(t), v(t), p^*(t), t) &= 0\,, \\
\dot{p}^* &= -\partial_x \mathcal{H}|_{x=x^*}\,, \\
\dot{x}^* &= \partial_p \mathcal{H}|_{p=p^*}\,,
\end{aligned} \tag{26}
$$

where the co-state vector $p(t)$ is the canonical momentum in Hamiltonian mechanics. In a similar fashion, the quantum version of Hamilton's equation of motion [33] can be derived in the stochastic setting. It follows from the saddle-point of Pavon's functionals introduced in the previous Section [32] within stochastic optimal control theory with respect to the two competing players $v = (v(t))_{t \in [t_0, t_1]}$ and $u = (u(t))_{t \in [t_0, t_1]}$, eventually considered as feedback controls $\tilde{v}(x, t)$ and $\tilde{u}(x, t)$ associated with finite-energy diffusions. One can either search for a Nash equilibrium [36] for the two variational principles or solve the complex variational problem by introducing a complex velocity, the so-called *quantum velocity* $v_q = v - \mathrm{i}u$,

$$J[v_q] = \mathrm{E}\left[\int_{t_0}^{t_1} \mathcal{L}(t, X(t), v_q(t))\mathrm{d}t + \Phi_1(X(t_1))\right]. \tag{27}$$

subject to the control equation

$$\mathrm{d}X(t) = v_q(t)\mathrm{d}t + \frac{1}{2}\sigma\big((1+\mathrm{i})\mathrm{d}W(t) + (1-\mathrm{i})\mathrm{d}W_*(t)\big), \quad X(0) = x_0. \tag{28}$$

This SDE can be understood within the theory of backward doubly SDEs. The final (or if needed initial) value should have the form $\Phi_1(X(t_1)) = -\mathrm{i}\hbar R(t_1, X(t_1)) + S(t_1, X(t_1)) = -\mathrm{i}\hbar R_1(X(t_1)) + S_1(X(t_1))$ with differentiable functions $R(\cdot, x(\cdot))$, $S(\cdot, x(\cdot))$ in $x$. An associated stochastic optimal control Hamiltonian to (27) [37] can be defined as

$$\mathcal{H}(X, v_q, P, Q, t) = -\mathcal{L}(X, v_q, t) + Pv_q - \frac{1+i}{2}\sigma Q, \tag{29}$$

where adapted complex stochastic costate variables $P(t)$ and $Q(t)$ occur. The additional stochastic processes $P$ and $Q$ now satisfy a backward SDE, e.g., given in [37] ($\mathrm{d}t > 0$);

$$\mathrm{d}P(t) = -\partial_x\mathcal{H}\,\mathrm{d}t + Q(t)\,\mathrm{d}W_*(t), \quad P(t_1) = \partial_x\phi_1(X(t_1)). \tag{30}$$

This is the adjoint equation backward in time to the constraint (28). Note that for a non-euclidean metric space, generally a (pseudo-)Riemannian manifold, $\sigma$ depends on the metric and hence, may depend on generalized coordinates, which is not considered here. If the system's potential does not depend on the velocity of the particle, then $\partial_x\mathcal{H} = \partial_xV$. The SDE for $P(t)$ is backward in time by definition of the Hamilton function, i.e., one could also introduce a co-state process $\underline{P}(t)$ that satisfies a forward SDE, depending on initial or final conditions. Finding critical points of the Hamiltonian corresponds to finding the roots of the complex functional w. r. t. $v_q$, leading to

$$P(t) = mv_q(t). \tag{31}$$

The Madelung equations for the velocities $v$ and $u$ are recovered by applying the Itô-formula to $\tilde{P}(X(t), t) = m\tilde{v}_q(X(t), t) = P(t)$ in (30) and comparing the drift terms. This means the dynamical SDEs correspond to Nelson's Newton-like law for the mean acceleration. The stochastic process $Q$ can be calculated following Itô's formula

$$Q(t) = \sigma\partial_x\tilde{v}_q(X(t), t). \tag{32}$$

Thus, Equations (30) with (31) and (28) describe the quantum system solely in terms of SDEs. The following system of coupled FBSDEs for a non-stationary system for the feedback controls [33]

$$\mathrm{d}X(t) = [\tilde{v}(X(t), t) + \tilde{u}(X(t), t)]\mathrm{d}t + \sigma\mathrm{d}W_f(t)$$
$$\mathrm{d}X(t) = [\tilde{v}(X(t), t) - \tilde{u}(X(t), t)]\mathrm{d}t + \sigma\mathrm{d}W_b(t) \tag{33}$$
$$m\mathrm{d}[\tilde{v}(X(t), t) + \tilde{u}(X(t), t)] = -\partial_xV(X(t), t)\mathrm{d}t + \sigma\partial_x[\tilde{v}(X(t), t) + \tilde{u}(X(t), t)]\mathrm{d}W_b(t)$$

are called the quantum Hamilton equations, where the imaginary and real part in (30) were used for the real momentum $m[\tilde{v}(X(t), t) + \tilde{u}(X(t), t)]$. Formally, this set of kinematic and dynamical equations describes quantum systems that can be related through the Itô formula to the Madelung equations, which again are the generalized version of the Schrödinger equation, and thus allow a broader case of solutions in general. For the ground state of stationary problems, i.e., node-free wavefunction, they are equivalent. However, as shown in [34,39], the Schrödinger equation is a simplification of these equations, thus, in general, the Madelung equations allow more solutions than the Schrödinger equation. The quantum Hamilton equations at hand yield the stable ground state solution, since they are the **unique** critical solution of the posed variational problem. The excited states of a bound spectrum are determined iteratively with a supersymmetric procedure exemplified in Section 4.1 or Section 4.3.

The similarity of (33) and (30) to Hamilton's equation of motion can be seen by taking $\hbar/m \to 0$, where the osmotic velocity $u$ vanishes and $P(t) = m\tilde{v}(X(t), t)$,

$$
\begin{aligned}
\frac{\mathrm{d}}{\mathrm{d}t}x(t) &= \frac{P(t)}{m} \\
\frac{\mathrm{d}}{\mathrm{d}t}P(t) &= -\frac{\partial}{\partial x}V(x, t)
\end{aligned}
\tag{34}
$$

These are the classical equations of motion. They predict the same outcomes as, e.g., the Hamilton–Jacobi theory, but from a different point of view. The conceptual similarities between classical and quantum mechanics on the basis of their mathematical formulations can be drawn, e.g., for the Hamilton–Jacobi equation and the Schrödinger equation, Hamilton's variational principle and Pavon's quantum Hamilton principle or Hamilton's equations of motion and their stochastic generalization. All of them can be solved independently in their framework while being, at least to some extend, equivalent in the case of quantum systems.

The quantum Hamilton equations are fully coupled forward backward SDEs in general which makes them hard to solve analytically and numerically compared to standard non-relativistic techniques. However, some examples regarding stationary QHEs are given in the upcoming sections.

### 4.1. Harmonic Oscillator

The problem of the stationary one-dimensional harmonic oscillator with potential $V_0(x) = \frac{m\omega^2}{2}x^2$ leads to QHEs [33]

$$
\begin{aligned}
\mathrm{d}X(t) &= \tilde{u}(X(t))\mathrm{d}t + \sigma \mathrm{d}W_f \\
\mathrm{d}X(t) &= -\tilde{u}(X(t))\mathrm{d}t + \sigma \mathrm{d}W_b \\
\mathrm{d}\tilde{u}(X(t)) &= \omega^2 X(t)\mathrm{d}t + \sqrt{\frac{\hbar}{m}}\partial_x \tilde{u}(X(t))\mathrm{d}W_b,
\end{aligned}
\tag{35}
$$

where the current velocity $v$ vanishes. Taking the expectation values of (35), the averages $E[X(t)]$ and $E[u(t)]$ follow classical paths, i.e., if $(X_0(t), u_0(t))$ is a solution then $(X_0(t) + X_{\text{classical}}(t), u_0(t) + u_{\text{classical}}(t))$ is a solution to the Problem similar to the coherent states for the quantum harmonic oscillator. These coupled forward backward SDEs can be solved numerically [33] or analytically with the help of the Itô formula, leading to the ground state solution $\tilde{u}_0(x) = -\omega x$. The stochastic process $X_0 = (X_0(t))$ corresponding to $\tilde{u}_0(x)$ is the ground state process with a stationary distribution of $\rho_0(x) = Ne^{-\frac{m\omega}{\hbar}x^2}$, where $N$ is the normalization constant. Thus, it is the Gaussian ground state probability corresponding to the wavefunction for the harmonic oscillator according to Born's rule and the energy expectation value is

$$
\mathrm{E}\left[\frac{m}{2}\tilde{u}_0^2(x) + V_0(x)\right] = \frac{1}{2}\hbar\omega.
\tag{36}
$$

As mentioned in Section 3, the excited states are not given directly as solutions of the optimal control formulation, as the ground state yields the one unique optimal control. However, the complete bound spectrum can be determined using the supersymmetric construction [40,41], which dates back to a concept in mathematics for a special type of differential equation, see e.g., [42,43]. This well-known formalism allows to state partner Hamiltonians through adjusting the potentials, leading to the quantized raising (or lowering) of the mean energy. In terms of the QHEs in stochastic mechanics, this is done by adjusting the partner potential $V_n = V_{n-1} - \hbar \partial_x \tilde{u}_{n-1}^0$ of the $n$-th energy level iteratively, where $\tilde{u}_{n-1}^0$ is the ground state, indicated by the upper index 0, of the $(n-1)$-th partner potential, indicated by the lower index, e.g., the first partner potential calculated from the ground state $\tilde{u}_0^0 = \tilde{u}_0 = -\omega x$ is just shifted by a constant $V_1 = V_0 + \hbar \omega$, i.e., the averaged energy is shifted by $\hbar \omega$. From this, it follows that the ground state solution to the partner potential $V_1$ equals the ground state solution of $V_0$, cf. (35), the $n$-th partner potential is $V_n = V_0 + n\hbar \omega$, and the corresponding mean energy is

$$\mathrm{E}\left[\frac{m}{2}\tilde{u}_0^2(x) + V(x)\right] = \left(\frac{1}{2} + n\right)\hbar \omega \,. \tag{37}$$

Note that the ground state solutions to the partner potential are not the excited states since they are node-free. The actual excited states of the harmonic oscillator (including nodes in the probability distribution) can be calculated iteratively based on the osmotic velocities as follows [44]

$$\tilde{u}_0^n(x) = \tilde{u}_0^{n-1}(x) + \frac{\hbar}{m}\partial_x \ln\left[\tilde{u}_0^0(x) + \tilde{u}_0^{n-1}(x)\right] \tag{38}$$

without the use of the wave function. Thus, the energy spectrum and the bound states can be determined completely with the quantum Hamilton equations.

### 4.2. Quantum Tunneling

"How long does it take to tunnel through a barrier?" This is an old question in quantum mechanics [45] that acquired new urgency with the emergence of attosecond experiments in recent years. Several papers have been published claiming instantaneous tunneling, see, e.g., [46,47], and finite time tunneling [48]. The lack of a time operator in standard quantum mechanics and the non-local behavior of the wave function leaves room for various definitions of tunneling times, e.g., there is the Wigner (delay) time based on the wave packet's peak under the barrier and the related phase shift [49], the traversal time [50] relying on the transmission analysis of the particle through a static barrier or the Larmor time which depends on the precession angle due to a magnetic field in the barrier region [51,52].

In the stochastic picture of quantum mechanics, the "tunneling" time can be directly obtained from the history of the stochastic process in real-time, i.e., the time spent 'interacting with' the barrier can be calculated for each path. The ensemble of these paths allows calculating a probability distribution for the tunneling times. The "tunneling" itself in Nelson mechanics is possible because the system is open, thus the energy of the conservative diffusion is not fixed but fluctuating. This energy fluctuation naturally entails that a particle can overcome any finite potential barrier.

Hence, in this picture, one can calculate an average time that is needed to cross a barrier instead of traversing a 'classically' forbidden region. Let us consider as an example a stationary system of the quartic double-well potential $V(x) = \frac{V_0}{a^4}(x^2 - a^2)^2$ with barrier height $V_0$ and the location of the minima $\pm a$ [41]. With the quantum Hamilton equations, the ground state osmotic velocity $\tilde{u}_0(x)$ and the excited states can be numerically obtained. This allows simulating the time a particle needs, starting at position $x$ located in a well, to arrive at some defined exit point $x_t$ after passing the barrier. In Figure 1, a sample path is shown for the stationary ground state of the double-well for $V_0 = 2$ and $a = 1.5$. It shows

the diffusive motion and transition between the two wells. The stochastic energy, which is not the average ensemble energy, shows that the "tunneling" in the stochastic picture is due to energy fluctuations. This diffusive motion is similar to thermally activated crossing of a barrier in a potential $V_K(x)$ in Kramer's theory. Here, $\ln \frac{1}{\rho_0(x)}$ plays the role of the diffusion potential with $\tilde{u}_0(x) \propto -\nabla \ln \frac{1}{\rho_0(x)}$, see the r.h.s. in Figure 1.

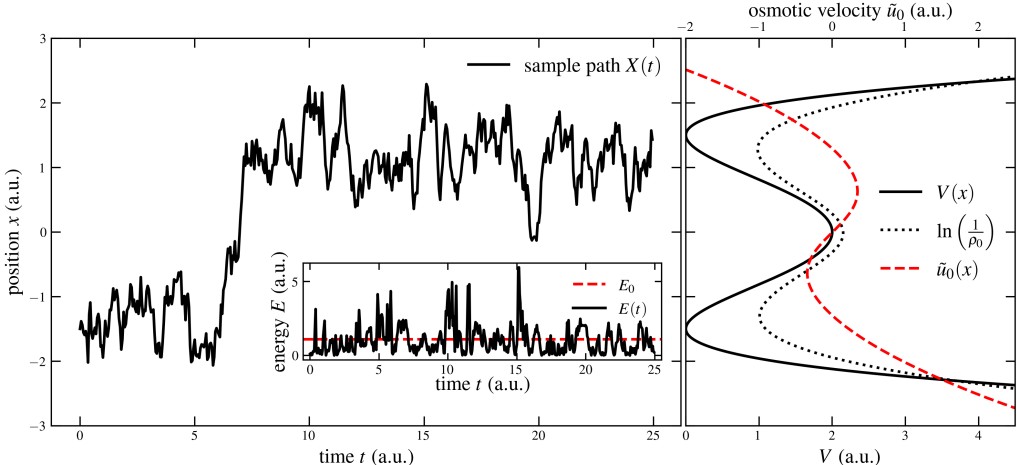

**Figure 1.** The left figure shows a sample path of the stochastic process $X$ in the ground state of the double-well potential driven by fluctuations, while the inset displays its stochastic energy $E(t) = \frac{m}{2}\tilde{u}^2(X(t)) + V(X(t))$ (solid black) in comparison to the mean ground state energy $E_0$ (dashed red). The graphic on the right shows the corresponding double-well potential $V(x)$ (solid black) with parameters $V_0 = 2$ and $a = 1.5$ and the osmotic velocity $\tilde{u}_0(x)$ (dashed red). The dotted black line depicts the 'diffusive potential' according to Kramers' theory $\ln \frac{1}{\rho_0(x)} \propto V_K(x)$ (dotted black) based on the ground state probability $\rho_0(x)$ of the system.

The first passage time starting from position $x$ can be averaged, resulting in a mean first passage time $t_m$ for a stationary problem [11]

$$t_m(x) = \frac{2m}{\hbar} \int_x^{x_t} \frac{dx'}{\rho_0(x')} \int_{-\infty}^{x'} \rho_0(x'') dx'' , \tag{39}$$

where the probability density can be calculated from the osmotic velocity $\ln \rho_0(x) = 2m/\hbar \int_{-\infty}^x \tilde{u}_0(x') dx'$. The average over all starting points $x$ of the mean first passage time $\langle t_m(x) \rangle$ can predict the so-called energy splitting [41]

$$\Delta E := E_1 - E_0 = c \frac{\hbar \pi}{\langle t_m \rangle} , \tag{40}$$

where $c \approx \frac{2}{\pi}$ is a constant number independent of the barrier parameters. $E_0, E_1$ are the expected energies of the ground and first excited state, respectively. The energy splitting for $V_0 = 2$, $a = 1.5$ and numerical results are shown in Figure 2. The inset shows that the prediction of the energy splitting within the stochastic picture is better compared to the instanton approach suggested in [53], where the splitting due to tunnel effects is approximated starting from the harmonic ground state. It agrees with the exact solution as long as the lowest state energies are smaller than the barrier height.

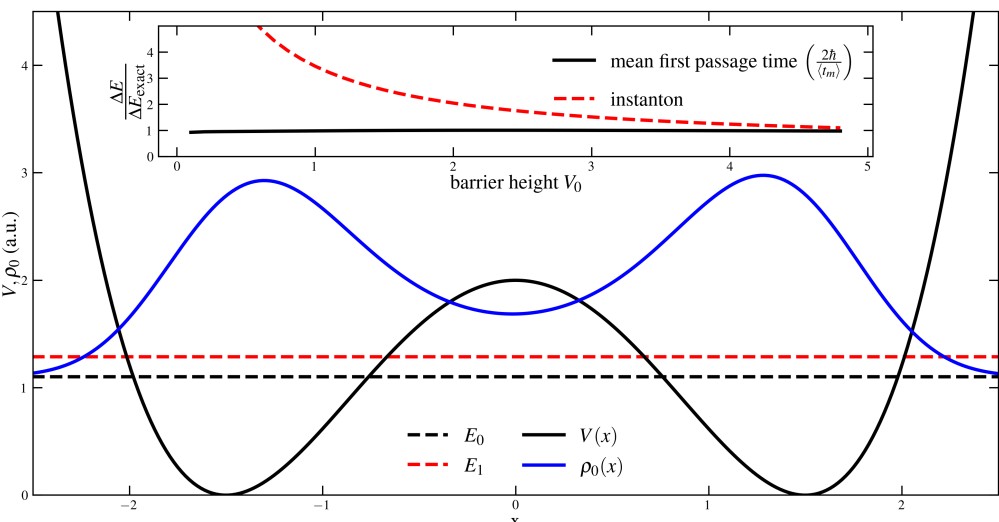

**Figure 2.** The big figure shows the ground state probability $\rho_0$ (solid blue), the first two mean energies $E_0$ (dashed black) and $E_1$ (dashed red) and the potential $V$ (solid black). The inset compares the ratio of predicted and exact energy splitting $\Delta E = E_1 - E_0$ of the mean first passage time (solid black) and an instanton approximation [53] (dashed red) depending on the barrier height $V_0$.

### 4.3. Hydrogen Atom

The quantum mechanical hydrogen atom is a two-body system, where proton and electron with opposite charges $e$ and $-e$ interact through the Coulomb potential $V(r) = -\frac{e_0^2}{r}$ where $e_0^2 = \frac{e^2}{4\pi\epsilon_0}$. In analogy to the Kepler problem in classical mechanics, one can make use of the systems symmetries: (1) translational and (2) rotational symmetry. The first leads to the conservation of the center of mass momentum under expectation, i.e., a free $3d$ Brownian motion or in terms of the wave function a plane wave. The relative coordinate between nucleus and electron $\boldsymbol{R}$ is a stochastic process and can be treated separately from the center of mass, leading to stationary stochastic differential equations [44]

$$\mathrm{d}\boldsymbol{R}(t) = \tilde{v}_q(\boldsymbol{R}(t))\mathrm{d}t + \sqrt{\frac{\hbar}{\mu}}\,\mathrm{d}\boldsymbol{W}_f(t)$$

$$\mu\mathrm{d}\tilde{v}_q(\boldsymbol{R}(t)) = \frac{e^2}{4\pi\epsilon_0 R^3(t)}\boldsymbol{R}(t)\mathrm{d}t + \sqrt{\frac{\hbar}{\mu}}\nabla_r\tilde{v}_q(\boldsymbol{R}(t))\,\mathrm{d}\boldsymbol{W}_b(t)\,,$$

(41)

where $\mu$ is the reduced mass. This is a Kepler-like system since under expectation

$$\mathrm{dE}[\boldsymbol{R}] = \mathrm{E}[\tilde{v}_q]\mathrm{d}t$$

$$\mu\mathrm{dE}[\tilde{v}_q] = \mathrm{E}\left[\frac{e_0^2}{R^3}\boldsymbol{R}\right]\mathrm{d}t\,,$$

(42)

which corresponds to Ehrenfest's theorem stating that the time derivative of the expectation values of the position and momentum operators obey the corresponding classical equations of motion. In classical mechanics, the proton would 'trap' the electron for zero angular momentum. That is where the $3d$ fluctuations play a role in this stable quantum state. To make this more clear, the isotropy of the system allows to further reduce the dimension of the problem, by changing to spherical coordinates where the angular part can be separated from the radial part [44]. In the stationary case, the current velocity $v$ vanishes in the ground

state and contributes only to the *z*-component of the angular momentum for excited states. This leads to quantum Hamilton equations for the radial part of the osmotic velocity

$$dR(t) = \left( \tilde{u}_r(R(t)) + \frac{\hbar}{\mu\, R(t)} \right) dt + \sqrt{\frac{\hbar}{\mu}} dW_f(t)$$

$$d\tilde{u}_r(R(t)) = \frac{1}{\mu R^2(t)} \left( e_0^2 + \hbar\tilde{u}_r(R(t)) + \frac{E\left[\tilde{L}^2\right]}{\mu R(t)} \right) dt + \sqrt{\frac{\hbar}{\mu}} \partial_r \tilde{u}_r(R(t))\, dW_b(t)\,. \tag{43}$$

where $\tilde{u}_r = \tilde{\boldsymbol{u}} \cdot \frac{\boldsymbol{R}}{R}$ is the radial projection of the osmotic velocity and $\tilde{\boldsymbol{L}} = \mu(\boldsymbol{R} \times (\tilde{\boldsymbol{v}} + \tilde{\boldsymbol{u}}))$ is the stochastic angular momentum for the stationary problem. The additional drift terms in (43) occur due to the transformation of variables, e.g., $\hbar/\mu R$ is a probabilistic drift due to the fluctuations in three dimensions, which try to push the particle away from the center. They follow from the non-vanishing variance of the stochastic process concerning the position in the mean square limit.

The solution to (43) for the ground state with zero mean angular momentum is isotropic, namely $\tilde{u}_r = -\frac{\hbar}{a_0\mu}$. This corresponds to the s orbital in which the electron moves diffusively in the ground state around the proton. The most probable distance between electron and proton is the Bohr radius $a_0$, but different from the Bohr model, there is no Kepler-like elliptic motion of the electron. This is due to the lack of a mean angular velocity. Note that the classical Kepler-problem should be recovered in the Bohr correspondence limit, which was shown for the Nelson diffusion in two dimensions [14]. The excited states can again be obtained with the help of partner potentials, see [44]. From the QHEs alone, there is no restriction on the angular momentum to be quantized since they lack the same additional quantization condition as the Madelung equations mentioned by Wallström [39]. A possible solution to this problem may be found in stochastic electrodynamics where the stochasticity is due to the zero-point radiation field [54].

The supersymmetric procedure for the QHEs leads to a constant expectation value of the square of the angular momentum in (43)

$$E\left[\tilde{L}^2\right] = -\mu^2\tilde{u}_\vartheta^2 - \hbar\mu \cot\vartheta\, \tilde{u}_\vartheta + \frac{\mu^2}{\sin^2\vartheta} \tilde{v}_\varphi^2 - \hbar\mu\left[\partial_\vartheta\tilde{u}_\vartheta\right] = \hbar l(l+1)\,. \tag{44}$$

Here, $\tilde{u}_\vartheta = R\tilde{\boldsymbol{u}} \cdot \hat{\boldsymbol{\vartheta}}$ and $\mu\tilde{v}_\varphi = \tilde{L}_z$ are the angular osmotic velocity and the constant *z* component of the angular momentum, respectively, and *l* is the orbital quantum number in the solution of the Schrödinger equation. The radial osmotic velocities for the states corresponding to the energy level *n*, $l = n-1$ read $\tilde{u}_r^{n,l=n-1} = -\frac{\hbar}{(l+1)\mu a_0} + \frac{\hbar l}{\mu R}$ while the osmotic angular velocity depends on the value of the angular momentum $\tilde{v}_\varphi$, which in turn corresponds to the magnetic quantum number in standard quantum mechanics.

## 5. Conclusions

In this short review, we have presented the state of the art of the stochastic mechanics' formulation of quantum mechanics, an endeavor based on Nelson's seminal paper from 1966. We have shown that this stochastic theory of quantum mechanics by now exists in a Newtonian formulation (Nelson's original work), a Lagrangian formulation, a Hamiltonian formulation (quantum Hamilton equations which can be used to solve a quantum problem without touching the Schrödinger equation), and, of course, Schrödinger's original Hamilton–Jacobi formulation of quantum mechanics. The theory thus parallels the structure of classical analytical mechanics, i.e., we have a quantum analytical mechanics at our disposal. Quantum analytical mechanics is the description of continuous but not differentiable trajectories in configuration space in the same way as classical analytical mechanics is the description of twice continuously differentiable trajectories in configuration space.

To date, this has been completely established for the position space of an N-particle system, for the internal degree of freedom called spin, the quantum Hamilton equation formulation is still missing, but currently under development in our group.

The underlying mathematics is that of conservative diffusion processes on Riemannian manifolds. Interesting questions arise when one considers the transferal of this approach to the Minkowski space-time of special relativity. Instead of Gaussian diffusion processes, one will need to consider Lévy processes [55,56], i.e., jump diffusion processes or pure jump processes [57]. The mathematical theory of stochastic optimal control underlying the derivation of quantum Hamilton equations is applicable to these cases [36], but all this still has to be worked out. Another suggestion that has been put forward in this context is to parameterize the world line of a particle in $M^4$ via the proper time to generalize Nelson's idea [58]. Finally, as the stochastic formulation of quantum mechanics is already cast in the language of motions on manifolds, a unification of quantum mechanics and general relativity via this route seems most promising [59].

**Author Contributions:** Conceptualization, M.B. and W.P.; writing—original draft preparation, M.B. and W.P.; writing—review and editing, M.B.; visualization, M.B.; supervision, W.P.; project administration, W.P.; funding acquisition, W.P. All authors have read and agreed to the published version of the manuscript.

**Funding:** This research received no external funding.

**Institutional Review Board Statement:** Not applicable.

**Informed Consent Statement:** Not applicable.

**Conflicts of Interest:** The authors declare no conflict of interest.

## Abbreviations

The following abbreviations are used in this manuscript:

| | |
|---|---|
| QHEs | quantum Hamilton equations |
| SDE | stochastic differential equation |
| FBSDE | forward backward stochastic differential equation |

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
