# Peer review of "On the Stochastic Mechanics Foundation of Quantum Mechanics"

_universe, doi:10.3390/universe7060166_

Round 1

Reviewer 1 Report

This is a well done review of stochastic mechanics, which may be of great help to those interested in the subject and with sufficient mathematical knowledge of stochastic methods.

Being a review, the question of originality does not apply to it.

It would benefit from minor language editing.

Author Response

Thank you for reviewing the manuscript.
We did the required language editing.

Reviewer 2 Report

The motivations for this work are valid, relevant, and clearly expressed in the Introduction. The manuscript is well written, using good English. However, the authors could emphasize the novelty of the study. This may be done by minor adjustments in the introduction.

The presented literature is comprehensive and it covers publications from year 1926 until this year. Although, the role of latest papers on the stochastic quantum mechanics and recent debates e.g. Rizzi, A. Does the PBR Theorem Rule out a Statistical Understanding of QM? Foundations of Physics, Volume 48 (2018) may be clarified. This can be done for example by clarifying the scope and limits of the recent study and/or elaborate the conclusions.

These considerations do not preclude publication. They can be taken into account with minor changes.

Author Response

We are grateful for your helpful comments and suggestions.
Some necessary minor language editing was done.

However, the authors could emphasize the novelty of the study. This may be done by minor adjustments in the introduction.

Regarding the novelty, the article is supposed to be a review. Thus, it aims
to give an overview of the developments in the field of stochastic mechanics with a focus on variational principles and especially the recently derived quantum Hamilton equations. 

Although, the role of latest papers on the stochastic quantum mechanics and
recent debates e.g. Rizzi, A. Does the PBR Theorem Rule out a Statistical
Understanding of QM? Foundations of Physics, Volume 48 (2018) may be
clarified.

The PBR theorem is not in contradiction with stochastic mechanics. The "ontic" wave-functions all carry the
the corresponding osmotic or diffusion potential which occurs in the amplitude
of the wave-function. This potential is in general not separable for multi
particle systems. We decided to not include this or similar 'debates' into the review
since it did not seem to fit in the context of this short review.

This can be done for example by clarifying the scope and limits of the recent
study and/or elaborate the conclusions."

At the end of section 4 and 4.3 some remarks regarding the limited solvability
and practicability of the stochastic approach were added. 

Reviewer 3 Report

According to its Conclusions, the submitted paper is a "short review ... [of] ... the state of the art of the stochastic mechanics formulation of quantum mechanics," a body of theories that is by now well developed, but undeservedly neglected by a scientific audience often focused on other problems deemed more important. For this reason the article is welcome, its aims are commendable and it deserves to be favorably considered for publication. The paper ends also in the Section 4 with three interesting applications of the "quantum Hamilton equations" (33) that seem however essentially drawn from a few previous papers by the authors themselves (Ref.s [35], [42], [43])

While however a more comprehensive review would arguably be beyond the scope of this concise article, it must be emphasized that the main reason for the neglect in which the stochastic mechanics has fallen is probably the lack of a widespread acquaintance of its methods and of its physical and mathematical bases. As a conequence a paper devoted to rescue this theory from its seeming marginality -- if not stikingly new in its content -- should at least be formally perspicuous and conceptually convincing: and in these respects the submitted paper can be improved.

It is unfortunate, in particular, that to represent processes and random variables the authors chose an outdated notation that -- while still popular among physicists -- turns out to be a source of obscurities and misunderstandings. They in fact indicate both the random variables (stochastic processes) and their values (sample paths) with lowercase latin letters (x, y, x(t), y(t) ...) instead of discriminating them by adopting different symbols: it is usual indeed today to indicate the random quantities with capital latin letters (X, Y, X(t), Y(t) ...) or their greek counterparts (ξ, η, ξ(t), η(t) ..., as it is often done in the Eastern European tradition), and their values/paths with the lowercase letters. 

This could appear as a minor point, and it would indeed be such (and not worth mentioning) if the authors paid particular attention to both the wording and the use of their notation: but -- in the opinion of this referee -- this seems not to be the case, and it is far from enough to add at the line 81/82 the passing remark "Note that x or x(t) is used as variable or stochastic process depending on the context". Let us look for instance at the Secion 2 where the basics of the theory are briefly provided: 

line 76: x(t) is not a "Brownian motion" but rather a diffusion process powered by the background Brownian motion Wf(t) according to the subsequent SDE.

lines 76/82: please explain what a "conservative" diffusion is: the wording is right, but certainly enigmatic to many.

eq.s (1) (2): please say somewhere that these (and subsequent) equations are particular Ito SDE's. In order to look as SDE's, however, these should be written rather as 

dx(t) = bf(x(t), t) dt + dWf(t) 

with the process x(t) frankly appearing even in bf(...) at the r.h.s. of the equation: since the authors do not adopt a different notation for the processes, simply write down bf(x,t) is misunderstood  as a mere numerical function, and the "equation" ... disappears

point 3 (line 91 and following): there is a disturbing confusion between the mean forward and backward derivatives with a conditioning w.r.t. the random variables x(t) (in this sense, in the eq.s (3) the operators Df and Db act on processes and produce processes), and  those with a conditioning w.r.t. the events x(t) = x (that of course would be more understandable if written as "X(t) = x" to tell apart random variables and their values) that produce mere functions: if Df and Db in eq. (3) are conditional w.r.t.  x(t) = x, they indeed produce the functions bf(x,t) and bb(x,t) as in eq. (5); but then it is difficult to understand how they can act in the "stochastic Newton law" (4) where only stochastic processes and not functions are to be seen. On the other hand, if in the author's understanding Dfx(t) and Dbx(t) are functions, how can they define the acceleration by applying again the same operators to functions and not to processes? Would not be better distinguish between the Nelson derivatives on processes, and the corresponding fluidodynamical "substantial" operators acting on functions? There are of course unambiguous formulations of all that, but this would require a notation and a care that the authors decided to skip in their quest for excessive succintness.

line 106: As the eq.s (6) and (7) are both presented as "Fokker-Planck equations", it would be advisable to say a few words about the fact that in (7) the diffusion coefficient has the "wrong sign" (it is negative, as it sould NOT be). A comment about the possible time inversions needed to mend this seeming anomaly would be welcome.  

eq. (10): As for the eq.s (1) and (2) the process x(t) should explicitly appear even in the r.h.s.'s in order to have a meaningful equation. This is even more indispensable here because, just two lines below, in eq. (6) v(x,t) is used as an ordinary function, while in (1) and (2) it should be understood as v(x(t), t) ... x(t) being a process

eq. (11): It would be nice to add that the hypothesis that the current velocity is curl-free (a non trivial assumption for processes in more than 1 dimension) is no longer needed and comes rather as a result in the variational formulation of Guerra and Morato (Ref [15])

eq. (12): This formula does not qualify as an Ito formula unless you replace x by x(t), adding also that dx(t) is a stochastic differential and that by consequence (dx(t))2 does not behave as an higher order infinitesimal, but it is rather of the first order in dt and must be kept into account

Similar remarks could be extended even to the subsequent sections of the paper, but we will not indulge in a detailed listing of sort being sure instead to have been rather clear with the previous suggestions. As a matter of fact (at least in its elementary, early formulation) the SM should be clearly explained because it is not well known and there are several biased preconceptions against it. A clumsy formulation is therefore not welcome: maybe it would be better just to expose the theory in words (without many inaccurate equations) with a comprehensive list of references, as done on the other hand for most of the exposition. Since nothing is really proved here, the mathematical formulation can be kept at the minumun needed to establish a proper notation for later convenience along the paper

In conclusion the authors should be more careful about their statements and their notation. On the other hand in this paper there are neither proofs nor new theoretical results to speak about, and hence the redaction perspicuity of this "short review" should be deemed paramount to its apprisal for publication. It is therefore warmily suggested that the authors pay some extra care to thier notation and wording in order to produce a paper helpful to redeem the stochastic mechanics from its present neglect

PS: Typos

line 30: "canon ball" should be "cannon ball"

line 170: "is prove" should be "proves"

lines 152, 172: "Kopenhagen" in english better "Copenhagen"

Author Response

We are grateful for your helpful comments and suggestions
which we all tried to include in the revised version of the manuscript.
Let us detail our responses in the following.

"It is unfortunate, in particular, that to represent processes and random
variables the authors chose an outdated notation that -- while still popular
among physicists -- turns out to be a source of obscurities and
misunderstandings. They in fact indicate both the random variables (stochastic
processes) and their values (sample paths) with lowercase latin letters (x, y,
x(t), y(t) ...) instead of discriminating them by adopting different symbols:
it is usual indeed today to indicate the random quantities with capital latin
letters (X, Y, X(t), Y(t) ...) or their greek counterparts (ξ, η, ξ(t), η(t)
..., as it is often done in the Eastern European tradition), and their
values/paths with the lowercase letters. [...] 
Similar remarks could be extended even to the subsequent sections of the
paper, but we will not indulge in a detailed listing of sort being sure
instead to have been rather clear with the previous suggestions. As a matter
of fact (at least in its elementary, early formulation) the SM should be
clearly explained because it is not well known and there are several biased
preconceptions against it. A clumsy formulation is therefore not welcome:
maybe it would be better just to expose the theory in words (without many
inaccurate equations) with a comprehensive list of references, as done on the
other hand for most of the exposition. Since nothing is really proved here,
the mathematical formulation can be kept at the minumun needed to establish a
proper notation for later convenience along the paper." 

We decided to use the suggested notation, i. e. big letters for stochastic
quantities and small ones for their values. The optimal control section
follows the notation used in stochastic optimal control theory studies,
e. g. Oksendal 2014 [36]. 

"line 76: x(t) is not a "Brownian motion" but rather a diffusion process
powered by the background Brownian motion Wf(t) according to the subsequent
SDE." 

This has been adjusted.

"lines 76/82: please explain what a "conservative" diffusion is: the wording
is right, but certainly enigmatic to many." 

The conservative diffusion is now explained as non-dissipative/time-reversible diffusion.

"line 106: As the eq.s (6) and (7) are both presented as "Fokker-Planck
equations", it would be advisable to say a few words about the fact that in
(7) the diffusion coefficient has the "wrong sign" (it is negative, as it
sould NOT be). A comment about the possible time inversions needed to mend
this seeming anomaly would be welcome." 

A comment on the sign has been added.

"eq. (11): It would be nice to add that the hypothesis that the current
velocity is curl-free (a non trivial assumption for processes in more than 1
dimension) is no longer needed and comes rather as a result in the variational
formulation of Guerra and Morato (Ref [15])" 

This has been added.

"PS: Typos
line 30: "canon ball" should be "cannon ball"
line 170: "is prove" should be "proves"
lines 152, 172: "Kopenhagen" in english better "Copenhagen""

These typos have been removed and general language editing and spell checking has been done.